# Occupational Determinants of Leptospirosis among Urban Service Workers

**DOI:** 10.3390/ijerph17020427

**Published:** 2020-01-08

**Authors:** Azman Atil, Mohammad Saffree Jeffree, Syed Sharizman Syed Abdul Rahim, Mohd Rohaizat Hassan, Khamisah Awang Lukman, Kamruddin Ahmed

**Affiliations:** 1Department of Community and Family Medicine, Faculty of Medicine & Health Sciences, Universiti Malaysia Sabah, Kota Kinabalu 88400, Malaysia; azman.azmi.atil@ums.edu.my (A.A.); syedsharizman@ums.edu.my (S.S.S.A.R.); khamisah@ums.edu.my (K.A.L.); 2Department of Community Health, Universiti Kebangsaan Malaysia Medical Centre, Kuala Lumpur 56000, Malaysia; rohaizat@ppukm.ukm.edu.my; 3Borneo Medical and Health Research Centre, Faculty of Medicine & Health Sciences, Universiti Malaysia Sabah, Kota Kinabalu 88400, Sabah, Malaysia; ahmed@ums.edu.my

**Keywords:** leptospirosis, workers, occupational determinants, urban services

## Abstract

This study was carried out to determine the risk factors of leptospirosis infection among local urban service workers in Sabah. This is a cross-sectional study involving 394 workers in Kota Kinabalu City, Sabah, conducted from February to March 2017. Information on demography, occupational exposures and environmental factors was obtained by a modified validated questionnaire. Polymerase Chain Reaction (PCR) was used to determine the prevalence of positive leptospirae. The overall figure for positive leptospirae was 9.4% (95% CI: 6.8–12.8). Urban sweepers and lorry drivers made up the highest proportion of positive leptospirae respondents, contributing 15.5% and 9.4%, respectively. The significant risk factors for positive leptospirae were older age (*p*-value = 0.001), higher monthly salary (*p*-value = 0.039), longer duration of employment (*p*-value = 0.011) and working as an urban sweeper (*p*-value = 0.021). Leptospirae was prevalent among healthy urban service workers and relates to their working activities.

## 1. Introduction

Leptospirosis is a major public health concern worldwide and is considered one of the most widespread diseases of the past decade [1]. It is estimated that there are one million cases of leptospirosis worldwide each year, with an estimated death toll of about 58,900 [2]. Malaysia is known to be an endemic country for human leptospirosis [3]. The number of reported cases has risen dramatically since the Ministry of Health Malaysia highlighted leptospirosis as a notifiable disease in 2010, with reported cases increasing from 263 to 7806 in the duration of 10 years. The average annual incidence rate is 7.80 cases per 100,000 population, with an overall case fatality rate of 2.11% [4].

Leptospirosis is considered to be a primarily occupational disease because it is associated with people who have worked as miners, farmers, fishermen, veterinarians, military personnel, abattoir workers and sewer workers in the past [5]. Occupations that involve direct contact with soil, mud, or water put individuals at risk of contracting the disease [6]. Front-liners, such as the urban service workers, are likely to be exposed to the infection since their job description involves contact with water treatment, sewage, drains and drainage, sewers, garbage collection, and road sweeping [7].

The highest number of leptospirosis cases recorded in Sabah, Malaysia was 930 cases and 15 deaths in 2014, with the incidence rate for the disease amounting to 24.03 per 100,000 population [8]. Most of them were sporadic cases that might be related to occupational activities. At present, there are limited surveillance data that relate the reported cases of leptospirosis with types of occupation in Sabah. The lack of awareness programmes on leptospirosis infection among urban service workers, particularly in Sabah, might be contributing to the disease being currently widespread. Failure to overcome this issue will lead to considerable public health impact on the municipal organisation [9]. As part of the assessment of the total leptospirosis burden in Sabah, the purpose of this study is to determine the prevalence of leptospirosis and its associated risk factors among local urban service workers in Sabah.

## 2. Materials and Methods

This cross-sectional study was conducted among urban service workers in Kota Kinabalu City, Malaysian Borneo from February to March 2017. Kota Kinabalu is the capital and the largest city in Sabah, Malaysia, with a population of 452,058 people. The sample population was comprised of urban service workers, who are organised into four major occupational groups: urban sweeper, landscaper, garbage collector and lorry driver.

Inclusion criteria included working as a field urban service worker in Kota Kinabalu for at least 6 months. Universal sampling was carried out, and 394 urban service workers were purposively selected. Pre-screening with the Microscopic Agglutination Test (MAT) was conducted with subsequent Polymerase Chain Reaction (PCR) confirmation for those with seropositive MAT. Leptospirae positive was defined as positive on both MAT and PCR. Consenting participants were interviewed for information such as their socio-demographic factors (gender, age, ethnicity, marital status, household number, monthly salary and level of academic achievement), occupational factors (job category, working shift category, duration of employment, working while having a body wound, practising hand washing with soap at work, showering before going home, having food or drink while performing the job, having cigarettes while performing the job, animal contact in the workplace, encountering rats or rodents in the workplace and personal protective equipment (PPE) usage at work) and environmental factors (house status, house type, water source, toilet type, whether there was a river or paddy field near the house, household animal ownership, neighbours’ animal ownership, the presence of rats in the house, whether the household area was affected by flooding, accumulated garbage near the house and garbage disposal), as well as their recreational activities (swimming in rivers, gardening and fishing).

The questionnaire used was a modified version based on validated questions from previous studies [7]. The validation of the content in the questionnaires was achieved by cross-referencing and verification from experts. Informed written consents were obtained, and about 10 millilitres of blood was collected from each subject for PCR. PCR was chosen over MAT because it is more sensitive and specific for leptospirosis detection [10]. DNA was extracted from serum samples using a DNeasy blood and extraction kit (Qiagen, Hilden, Germany) according to the manufacturer’s instructions. To determine whether the serum samples of the urban service workers contained leptospirae, a nested PCR assay was performed [11].

The data were analysed by using Statistical Package for Social Sciences (SPSS) version 22. All continuous variables were described using mean and standard deviations, whereas frequencies and percentages were used for categorical variables. Univariable analysis using the independent sample t-test and simple logistic regression were carried out for continuous variables and categorical variables, respectively. Fisher’s exact test was used for variables with cell size less than 5. Variables with a *p*-value less than 0.2 were selected for multivariable analysis using multivariable logistic regression with the backward stepwise approach, in order to test independent factors for seropositive leptospirosis. The Hosmer–Lemeshow goodness of fit test and the receiver operator characteristic (ROC) curve were used in determining the fitness of the model. Statistically significant data were determined by a *p*-value of less than 0.05 with an adjusted odds ratio and 95% confidence interval. The prevalence of leptospirosis was computed using positive PCR over total samples and is presented as a percentage with 95% confidence interval (CI).

## 3. Results

### 3.1. Descriptive Analysis

Out of 394 respondents, 37 (9.4%) were positive for leptospirosis. Urban sweepers and lorry drivers make up the highest proportions of the positive PCR respondents, contributing 15.5% and 9.4%, respectively. The majority (79.4%) of the respondents were male, with a mean age of 42.6 (SD 9.6) years and a mean working experience of 14.9 (SD 11.6) years. Large proportions of the workers were Kadazan-Dusun-Murut (53.3%) in ethnicity and were married (81.0%). Table 1 describes the characteristics of urban service workers in Kota Kinabalu City.

### 3.2. Univariable and Multivariable Analysis

Table 2 shows the positive PCR leptospirosis distribution according to job category. Univariable analysis of factors associated with leptospirosis, using the independent sample t-test and simple logistic regression, is shown in Table 2.

The significant risk factors for positive leptospirae were older age (*p*-value = 0.001), higher monthly salary (*p*-value = 0.039), longer duration of employment (*p*-value = 0.011) and working as an urban sweeper (*p*-value = 0.021). Multivariable analysis using multiple logistic regressions revealed that only age was included in the final model (Table 3).

## 4. Discussion

Leptospirosis is commonly associated with high-risk occupations that involve contact with soil or water. The fundamental factors for the transmission of leptospirosis in humans are the presence of carrier animals, the suitability of the environment for leptospirae survival, and the interaction between human, animal and environment [12]. This study had a lower yield of PCR positive for leptospirosis among urban service workers, contrasting with findings in West Malaysia in Kelantan and Selangor, which recorded seropositive leptospirosis prevalence at 24.7% and 34.8%, respectively [6,9]. Healthy people living in the rural area of Sarawak had higher leptospirosis seroprevalence, which was recorded at 37.4% [13]. This finding explained that people in the rural area were also exposed to leptospirae during daily activities such as farming and hunting. However, the difference in prevalence might be due to the different method of detection, as all the studies mentioned above used MAT.

This study discovered that being in the older age group while working as an urban service worker in Kota Kinabalu was a risk factor for leptospirosis infection. In other words, the older the respondent is, the higher the risk of getting a leptospirosis infection. This might be because they have more time to be exposed and also because they might be chronic carriers. The result is comparable with findings in Brazil, where seropositive leptospirosis was significantly associated with increasing age [14]. This could be due to poor body response to infectious disease or the presence of comorbidity in the elderly.

This study also found out that having a higher monthly salary was significantly associated with leptospirosis. The result contrasts with a study done in Kuantan, Pahang in which lower income salary contributed to the inability to purchase PPE for leptospirosis prevention [15]. The difference might be due to the municipal centre’s ability to provide PPE to workers for free. Other socio-demographic factors such as gender, ethnicity, marital status, household number and level of academic achievement were found not to be significantly associated with seropositive leptospirosis. Similar findings were noted from a study done in Kelantan [7].

In this study, longer duration of employment was associated with contracting leptospirosis; this might be due to prolonged exposure. The magnitude of high-risk occupation and prolonged exposure to possible contaminated environments plays an important role in leptospirosis transmission [1,16]. Working as an urban sweeper is also significantly associated with leptospirosis. This finding is similar to studies done in Kelantan and Selangor [6,7]. Compared to the other job categories, urban sweepers had prolonged exposure to various mediums of environmental contaminants of leptospirosis such as garbage, soil and water. The only significant risk factor in the final logistic model of this study was the age of the urban service workers. The model explained that an increase of one year in age has a 7% (95% CI 3% to 11%) increase in the odds of having a leptospirosis infection.

This study could not find any association between environmental factors and the PCR positive leptospirosis infective respondents. Other studies in Kelantan and Fiji, for example, found significant association between living near water streams and leptospirosis infection [7,17]. This, and the presence of rats in houses, were significant factors in leptospirosis transmission [7,18,19]. However, the presence of other household or neighbourhood animals was not associated with leptospirosis infections [7,19].

## 5. Conclusions

Regular monitoring for this high-risk group of workers is mandatory by the respective authority in order to prevent unwanted morbidity and mortality. This study provides baseline data for public health personnel and policymakers to evaluate existing programmes of leptospirosis control among people in high-risk occupations in order to identify effective strategies and future programmes for behavioural change and subsequent reduction in the incidence of leptospirosis infection in Sabah as a whole.

## Figures and Tables

**Table 1 ijerph-17-00427-t001:** Characteristics of urban service workers in Kota Kinabalu, Sabah (*N* = 394).

Variables	Frequency (%)	Mean (SD)
Age (year)		42.6 (9.6)
Gender		
Male	313 (79.4)	
Female	81 (20.6)	
Ethnicity		
Kadazan-Dusun-Murut	210 (53.3)	
Brunei	66 (16.8)	
Bajau	55 (14.0)	
Rungus	45 (11.4)	
Others	18 (4.6)	
Marital status		
Married	319 (81.0)	
Single	52 (13.2)	
Widowed	23 (5.8)	
Household number		3 (2)
Monthly salary (RM)		1672.15 (614.52)
Level of academic achievement		
No formal education	11 (2.8)	
Primary school	85 (21.6)	
Secondary school	277 (70.3)	
College/university	21 (5.3)	
Job category		
Garbage collector	145 (36.8)	
Landscaper	99 (25.1)	
Urban sweeper	97 (24.6)	
Lorry driver	53 (13.5)	
Working shift category		
Daytime shift	341 (86.5)	
Night shift	53 (13.5)	
Duration of employment (year)		14.9 (11.6)

**Table 2 ijerph-17-00427-t002:** Factors associated with leptospirosis among 394 urban service workers in Kota Kinabalu, Sabah.

Associated Factor	Leptospirae Positive (*N* = 37)	Leptospirae Negative (*N* = 357)	Crude OR (95% CI)	*p*-Value
	No. (%)	No. (%)		
Age	47.8 (7.8) ^a^	42.1 (9.6) ^a^	5.71 (2.93, 8.49) ^b^	0.001
Gender				
Male	29 (9.3)	284 (90.7)	0.93 (0.41, 2.12)	0.867
Female ^c^	8 (9.9)	25 (90.1)		
Ethnicity				
KDM	20 (9.5)	190 (90.5)	1.03 (0.52, 2.04)	0.923
Non-KDM ^c^	17 (9.2)	167 (90.8)		
Marital status				
Married	29 (9.1)	290 (90.9)	0.84 (0.37, 1.91)	0.674
Single/widower ^c^	8 (10.7)	67 (89.3)		
Household number				
>3	14 (8.0)	160 (92.0)	0.75 (0.37, 1.50)	0.417
≤3 ^c^	23 (10.5)	197 (89.5)		
Level of academic achievement				
<Secondary school	12 (12.5)	84 (87.5)	1.56 (0.75, 3.24)	0.233
≥Secondary school ^c^	25 (8.4)	273 (91.6)		
Monthly salary (RM)	1876.1 (612.9) ^a^	1651.0 (611.7) ^a^	225.1 (11.81, 438.45) ^b^	0.039
**Occupational Factors**				
Duration of employment	19.7 (11.6) ^a^	14.4 (11.3) ^a^	5.29 (1.25, 9.33) ^b^	0.011
Job category				
Garbage collector				
Yes	9 (6.2)	31 (93.8)	1.92 (0.88, 4.18)	0.103
No ^c^	28 (11.2)	221 (88.8)		
Town sweeper				
Yes	15 (15.5)	82 (84.5)		
No ^c^	22 (7.4)	275 (92.6)	2.29 (1.13, 4.61)	0.021
Landscaper				
Yes	8 (8.1)	91 (91.9)		
No ^c^	29 (9.8)	266 (90.2)	0.81 (0.36, 1.83)	0.606
Lorry driver				
Yes	5 (9.4)	48 (90.6)		
No ^c^	32 (9.4)	309 (90.6)	1.01 (0.37, 2.71)	0.991
Working shift category				
Night shift	4 (7.5)	49 (92.5)	0.76 (0.26, 2.25)	0.802 ^d^
Daytime shift	33 (9.7)	308 (90.3)		
Working while having a body wound				
Yes	13 (7.6)	157 (92.4)	0.69 (0.34, 1.40)	0.303
No ^c^	24 (10.7)	200 (89.3)		
Practising hand washing with soap at work				
No	7 (13.2)	46 (86.8)	1.58 (0.65, 3.80)	0.309
Yes ^c^	30 (8.8)	311 (91.2)		
Taking a shower before going home				
No	31 (9.3)	302 (90.7)	0.94 (0.38, 2.36)	0.897
Yes ^c^	6 (9.8)	55 (90.2)		
Having food or drink while performing the job				
Yes	10 (8.3)	111 (91.7)	0.82 (0.38, 1.75)	0.61
No ^c^	27 (9.9)	246 (90.1)		
Having cigarettes while performing the job				
Yes	6 (8.7)	63 (91.3)	0.90 (0.36, 2.26)	0.828
No ^c^	31 (9.5)	294 (90.5)		
Animal contact in the workplace				
Yes	8 (12.3)	57 (87.7)	1.45 (0.63, 3.34)	0.38
No ^c^	29 (8.8)	300 (91.2)		
Encountering rats or rodents in the workplace				
Yes	34 (9.8)	312 (90.2)	1.64 (0.48, 5.54)	0.599 ^d^
No	3 (6.3)	45 (93.8)		
Personal protective equipment (PPE) usage at work				
No	3 (5.8)	49 (94.2)	0.56 (0.16, 1.88)	0.449 ^d^
Yes	34 (9.9)	308 (90.1)		
**Environmental Factors**				
House status				
Rented	14 (9.9)	127 (90.1)	1.10 (0.55, 2.22)	0.785
Owned ^c^	23 (9.1)	230 (90.9)		
House type				
Wooden	18 (7.9)	211 (92.1)	0.66 (0.33, 1.29)	0.222
Brick ^c^	19 (11.5)	146 (88.5)		
Water source				
Non-JBA	7 (10.1)	62 (89.9)	1.11 (0.47, 2.64)	0.813
JBA ^c^	30 (9.2)	295 (90.8)		
Toilet type				
Pit	3 (4.7)	61 (95.3)	0.43 (0.13, 1.44)	0.239 ^d^
Flush	34 (10.3)	296 (89.7)		
River near the house				
Yes	24 (9.7)	224 (90.3)	1.10 (0.54, 2.23)	0.799
No ^c^	13 (8.9)	133 (91.1)		
Paddy field near the house				
Yes	6 (8.7)	63 (91.3)	0.90 (0.36, 2.26)	0.828
No ^c^	31 (9.5)	294 (90.5)		
Household animal ownership				
Yes	18 (7.8)	212 (92.2)	0.65 (0.33, 1.28)	0.21
No ^c^	19 (11.6)	145 (88.4)		
Neighbours’ animal ownership				
Yes	25 (8.6)	267 (91.4)	0.70 (0.34, 1.46)	0.324
No ^c^	12 (11.8)	90 (88.2)		
Presence of rats in the house				
Yes	17 (8.6)	180 (91.4)	0.84 (0.42, 1.65)	0.605
No ^c^	20 (10.2)	177 (89.9)		
Household area affected by flooding				
Yes	2 (7.7)	24 (92.3)	0.79 (0.18, 3.50)	1.000 ^d^
No	35 (9.5)	333 (90.5)		
Accumulated garbage near the house				
Yes	12 (9.7)	112 (90.3)	1.05 (0.51, 2.17)	0.895
No ^c^	25 (9.3)	245 (90.7)		
Garbage disposal				
Non-public service	9 (7.3)	115 (92.7)	0.68 (0.31, 1.48)	0.328
Public service ^c^	28 (10.4)	242 (89.6)		
Recreational activities				
Swimming in rivers				
Yes	2 (14.3)	12 (85.7)	1.64 (0.35, 7.64)	0.631 ^d^
No	35 (9.2)	345 (90.8)		
Gardening				
Yes	16 (11.8)	120 (88.2)	1.51 (0.76, 2.99)	0.243
No ^c^	21 (8.1)	237 (91.9)		
Fishing				
Yes	4 (4.2)	91 (95.8)	0.35 (0.12, 1.03)	0.067 ^d^
No	33 (11.0)	266 (89.0)		

^a^ Mean (SD), ^b^ mean difference (95% CI), ^c^ reference group, ^d^ Fisher’s exact test. OR = odds ratio, CI = confidence interval, KDM = Kadazan-Dusun-Murut, RM = Ringgit Malaysia, JBA = Jabatan Bekalan Air.

**Table 3 ijerph-17-00427-t003:** Multivariable analysis for factors associated with leptospirosis among 394 urban service workers in Kota Kinabalu, Sabah.

Variable	Adjusted OR	95% CI	*p*-Value
Age	1.07	1.03, 1.11	0.001
Fishing	0.37	0.13, 1.07	0.067

OR = odds ratio, CI = confidence interval. Constant = −5.072. Hosmer-Lemeshow test *p*-value = 0.796. Receiver operating characteristics (ROC) curve = 0.70.

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
