# Peer review of "Occupational Determinants of Leptospirosis among Urban Service Workers"

_ijerph, 2020, doi:10.3390/ijerph17020427_

Round 1
Reviewer 1 Report
The paper presents interesting data and results. Tables are clearly presented. However, the authors should have their manuscript checked for English (native speaker) as lots of typos and minor errors can be found in the manuscript.
Author Response
Thank you so much for your time and effort in reviewing our manuscript. We greatly appreciate it.
We have taken into account your comment.
We have edited the language aspects and proofread the document.
Thank you again for highlighting the issues so we could further improve the manuscript.
Reviewer 2 Report
Thank you for the opportunity to review this manuscript. The manuscript described a study of leptospirosis infection in a high-risk group in Kota Kinabalu. The authors conclusions are that age and time of employment are associated with leptospirosis infection.
Overall the manuscript is well written and clearly presented. The conclusions are sensible. However, I have some concerns about the presentation of results and analysis which I have outlined below.
Major comments
Abstract
Age is normally presented as a risk factor rather than a protective factor. Authors should redo the calculation so that the odds ratios for respondents over 50 is presented. And the same for working greater than 15 years.
Methods
Line 55 Did the sample include all urban service workers or if not how were workers selected? Please describe
Line 69 Please confirm which PCR was used and describe including a brief description of the protocol and the primers
Line 73 The authors need to describe the methods of model building and how variables were selected for the multi variable model.
Results
Table 2 The table currently includes only one half of the information about each variable. Therefore, the reader only has half of the information. Please include the details of all the variable components. For example, include details for over 50, and male, etc.
Analysis
The sample size is small and there are some small cell sizes (<5). The logistic regression analysis with these small numbers may not be appropriate. The authors need to consider this in their analysis.
The reported variables from the multivariable model (age and time of employment) are highly correlated and should not be included in the model together. As mentioned above there is no description of how model building was done. It seems the authors have only included significant variables from the univariable analysis. The authors might consider relaxing the criteria to those variables with p- value less than 0.2 and then do a backward stepwise approach. Or choose variables that make biological sense. Given the number of positives they should not overfit the model and should not include more than 3-4 variables. Since age is a known confounder then the true relationship with other variables might be different?
As mentioned the significant variables should be written as risk factors not protective factors by reporting the Odd ratios for participants over 50 and those working more than 15 years.
I also suggest if authors have the information available that age and working time are included in the model as continuous variables rather than binary variables.
Minor comments
Introduction
Line 29 “The number of cases has risen dramatically” Please reword to “The number of reported cases has risen dramatically”. Otherwise this suggests the cases have increased where it is actually a change in reporting.
Line 36 Please reword “are likely to expose to the infection” to “are likely to be exposed to the infection”
Line 38 please define “galleries”. The use of this word is not obvious
Line 43 “There is also lacking of awareness programs implemented ….in contracting leptospirosis infection”. This sentence is hard to read and should be reworded.
Line 45 What do the authors mean by “organisation”
Methods
Line 66 “from a previous studies conducted” please reword to “from previous studies conducted”
Line 73 Please correct univariate to univariable and multivariate to multivariable throughout
Discussion
Line 149 “Leptospirosis was synonym with the high risk” needs to be reworded
Line 150 human should be humans
The authors have mentioned that PCR might not detect as many cases as MAT but need to discuss more the method of choice ie which method/primers?
Line 160 as stated above suggest that age is written as a risk factor not a protective factor. The authors do not mention that older age might be because of longer time to be exposed and that these people might be chronic carriers?
Line 169 as stated above suggest time of employment is written as a risk factor and not a protective factor
Author Response
We would like to convey our sincere gratitude to the reviewer for taking the time and effort to give such a detailed review.
We appreciate it greatly as it has put our paper at a new higher standard for publication.
We are attaching herewith the feedback reply for the comments given.
We hope to be considered for publication in this esteemed journal.

Round 2
Reviewer 2 Report
Review IJERPH 645042Thank you to the authors who have done a good job of addressing the points I raised in the first review.
I have one remaining concern about the manuscript.
It was not clear in the original version of this manuscript that the sampling frame was from urban workers prescreened for leptospirosis by MAT and I think this should be made obvious in the title and abstract since this changes the interpretation of the results. It also means this is no longer a cross sectional study. The authors should not report the prevalence when they are only reporting the results for a specific subset of the population.
If the authors want to keep the cross-sectional design then they may consider including the entire sample with the case definition as MAT and PCR positive and all other workers considered negative. I think the results would be more generalisable if this is done.
Minor
Line 42 this sentence is still very cumbersome
Suggest the following
“A lack of awareness programs on leptospirosis infection among the urban service workers particularly in Sabah might contribute to the disease being currently widespread.”
Line 49 this is not a cross sectional study in its current format since workers have been purposefully selected as MAT positive.
Line 55 Is it the authors plan to publish the MAT results? This would be very interesting data and I think the analysis might be more informative if these individuals were included.
Author Response
Response to comments: (Reviewer 2)
Line 42 this sentence is still very cumbersome
Suggest the following
“A lack of awareness programs on leptospirosis infection among the urban service workers particularly in Sabah might contribute to the disease being currently widespread.”
Changes has been made.
Line 49 this is not a cross sectional study in its current format since workers have been purposefully selected as MAT positive.
We kept it as a cross sectional study, and as suggested included the entire sample with leptospirosis case definition as MAT and PCR positive and all other workers considered negative.
Line 55 Is it the authors plan to publish the MAT results? This would be very interesting data and I think the analysis might be more informative if these individuals were included.
The MAT results will be published soon by different authors. This will next be an in depth study that includes molecular investigation.